# A Self-Coordinating Controller with Balance-Guiding Ability for Lower-Limb Rehabilitation Exoskeleton Robot

**DOI:** 10.3390/s23115311

**Published:** 2023-06-03

**Authors:** Li Qin, Houzhao Ji, Minghao Chen, Ke Wang

**Affiliations:** School of Electrical Engineering, Yanshan University, Qinhuangdao 066012, China; jilala6909@163.com (H.J.); chen11515130@163.com (M.C.); 18230296193@163.com (K.W.)

**Keywords:** lower-limb rehabilitation exoskeleton robotics, balance guiding, non-time-varying phase space, self-coordinated, adaptive trajectory generator

## Abstract

The restricted posture and unrestricted compliance brought by the controller during human–exoskeleton interaction (HEI) can cause patients to lose balance or even fall. In this article, a self-coordinated velocity vector (SCVV) double-layer controller with balance-guiding ability was developed for a lower-limb rehabilitation exoskeleton robot (LLRER). In the outer loop, an adaptive trajectory generator that follows the gait cycle was devised to generate a harmonious hip–knee reference trajectory on the non-time-varying (NTV) phase space. In the inner loop, velocity control was adopted. By searching the minimum L2 norm between the reference phase trajectory and the current configuration, the desired velocity vectors in which encouraged and corrected effects can be self-coordinated according to the L2 norm were obtained. In addition, the controller was simulated using an electromechanical coupling model, and relevant experiments were carried out with a self-developed exoskeleton device. Both simulations and experiments validated the effectiveness of the controller.

## 1. Introduction

As the population ages, the number of people with lower-extremity mobility problems continues to grow [1,2]. Due to the plasticity of the human neural system, the majority of lower-limb movement impairments caused by stroke, cerebral palsy, and other neurological diseases can be restored with the help of specific rehabilitation training [3,4]. LLRERs [5,6] have gradually surpassed therapists in popularity as a new method for lower-extremity rehabilitation training. They fall into two basic categories: suspended and freestanding. Suspended LLRERs, such as Lokomat [7], can use the suspension system to elevate the patient and keep them balanced. However, such LLRERs are typically huge, bulky in design, and expensive to train. The majority of patients prefer freestanding LLRERs over suspended LLRERs. The latter are compact, lightweight, and require no additional assistive devices such as Rewalk [8], HAL [9], or other similar devices, and they can be worn directly on the patient to assist with the majority of training scenarios. Due to the absence of an external balance assist device, the human–machine system’s balance is mostly dependent on the control strategy, which presents a design challenge.

For the balance control of freestanding LLRERs, some researchers try to solve this problem from the perspective of trajectory planning. By fitting human walking data, the early commercial exoskeleton Rewalk planned an offline trajectory that matched human gait balance characteristics. This offline planning scheme is overly reliant on the rationality of the fitted data, and its balance-assistance capability cannot be guaranteed because no two people walk the same way.

To maintain the dynamic balance of the human–machine system, researchers prefer online trajectory planning, and in [10,11,12,13,14], human physiological signals such as EMG and EEG signals were used to construct a human intention estimator that can be used to adjust the reference trajectory in real time based on the patient’s intention and to track the control. Both offline and online planning require the patient to follow a specified trajectory, which restricts the patient’s posture to some extent. This postural restriction can easily result in the exoskeleton producing forces that resist the user’s movement, resulting in patient imbalance [15,16].

In essence, the exoskeleton generates forces that interfere with the patient’s movement as a result of the non-compliant HEI process. Thus, when the patient interacts compliantly with the exoskeleton, the human–machine system’s balance performance can be significantly improved. To ensure the compliance of HEI, Modares et al. [17] proposed an HEI strategy that uses an optimal control framework to transform the goal of gait training into the LQR problem of finding the optimal impedance parameters of the exoskeleton by minimizing the kinematic conflict between the human and the exoskeleton. Therefore, the patient’s balance is guaranteed throughout the training process.

In addition, Zhang, T. et al. [18] combined the extrapolated centroid theory with adaptive admittance control [19] to generate a compliant HEI force, thus achieving the user’s gait balance in the sagittal plane and frontal plane. These solutions, which are based on soft interactions between humans and machines, can help patients expand their balancing options and increase their ability to maintain equilibrium. However, if compliance is pursued, the exoskeleton becomes increasingly subservient to the patient’s walking desires while drastically reducing gait correction, which is obviously dangerous for patients with an already abnormal gait.

Furthermore, an approach that has been used effectively is based on the human balance criterion, where the balance conformation is achieved directly by adjusting the patient’s joint angle [20,21,22]. These methods necessitate the addition of sensors and dynamic capture devices, impose stricter mechanical requirements, and are not always suitable for individuals with neurological disorders.

Building on the previously mentioned research, in this paper, an SCVV double-layer controller with balance-guiding ability is developed for the LLRER. It enables patients to walk steadily without crutches or other external balancing devices.

The main contributions of this article are as follows:

(1) Considering the harmony between joints, the concept of NTV phase space is introduced to solve the problem of asynchronous hip and knee joint trajectories in time-varying (TV) space.

(2) The concept of impedance control is combined in the outer loop of the controller, which ensures the compliance of the HEI process by minimizing the HEI torque.

(3) On the basis of the outer loop, an SCVV control inner loop is developed. By self-coordinating the proportion of encouraged and corrected velocity directions in real time, the current control target can be adjusted in real time, thereby providing balance guiding for the patient.

(4) In addition, through the combination of the inner and outer double loops of the controller, the influence of restricted posture and unrestricted compliance problems brought by the traditional controller on the balance is eliminated; meanwhile, the efficiency and quality of the lower-limb rehabilitation training are improved.

Finally, the controller was simulated using an electromechanical coupling model, and relevant experiments were carried out with a self-developed exoskeleton device. Both simulations and experiments validated the effectiveness of the controller.

This paper is organized as follows: Section 2 introduces the design of the SCVV controller based on the simple LLRER model in detail. Section 3 compares SCVV control, position control, and impedance control in simulation and proves the superiority of SCVV control in performance. Section 4 designs related experiments to further verify the performance of the SCVV control and compares it with the simulation results. Section 5 concludes this paper.

## 2. Design of SCVV Controller

The SCVV controller is intended to encourage the patient’s physiological gait, correct the patient’s non-physiological gait, and provide balance guiding during rehabilitation training. The controller is decoupled into two parts: the outer loop for adaptive gait generation and the inner loop for self-coordination balance guiding. In the outer loop of the controller, the authors designed an adaptive trajectory generator that can obtain reasonable trajectory parameters by introducing the concept of impedance control and combining the optimization principle, thereby generating harmonious hip–knee reference trajectories in an NTV phase space, which can be updated according to the gait cycle. In the inner loop, velocity control is adopted. By finding the minimum configuration error (in the sense of the L2 norm) between the reference phase trajectory and the current configuration, the desired velocity vectors through which the encouragement and correction effects are self-coordinated according to the configuration error are obtained, respectively. Through synthesizing the velocity vectors of the two action directions of encouragement and correction, the velocity reference vector that best matches the current state of the patient can be obtained, and the motor can be further controlled to track the velocity so that the patient can softly approach the training target while meeting the requirements of balance.

### 2.1. Outer Loop of the Controller

Since each person’s gait has a certain difference, a fixed standard gait trajectory curve cannot meet the needs of everyone. In order to improve the generalization ability of the reference gait trajectory, the trajectory parameters a and c are introduced, and the standard gait trajectory is parameterized. The results are as follows:(1)$$\varphi_{i}=a_{i}\varphi_{i,\mathrm{nom}}\left(\frac{t}{c_{i}}\right)i=1,2$$

$\varphi_{i,\mathrm{nom}}$ represents the nominal reference trajectory provided by the gait database, t represents the time period of a nominal reference trajectory, the $a_{i}$ parameter scales the amplitude of the reference trajectory, the $c_{i}$ parameter stretches the time and affects the gait cycle, i=1 represents the hip joint, and i=2 represents the knee joint. Since the hip and knee joint trajectory cycles of the same leg are the same, $c_{1}=c_{2}$. By changing the value of the trajectory parameter, different hip and knee joint trajectories can be generated based on the nominal reference trajectory. When initialized, all trajectory parameters are set to 1.

To improve training effectiveness, the LLRER training goal must be adjusted in response to the patient’s intention, ensuring that the patient participates actively in the training process. It has been confirmed [23,24] that the HEI torque partially reflects the patient’s intention, and that when the patient does not meet the current training goal, a greater HEI force is generated with the exoskeleton to change the training goal; when the current training goal is close to the patient’s expectation, the HEI torque decreases.

According to impedance control, when the robot interacts with the environment, there is an impedance relationship Z between the robotic end contact force $F_{\mathrm{end}}$ and the deviation e between the reference and the actual robotic end position.
(2)$$F_{\mathrm{end}}=Z(s)\cdot e$$
(3)$$Z(s)=K_{P}+K_{D}s$$
where $K_{P}$ is the proportionality coefficient and $K_{D}$ is the integral coefficient.

Because the patient can be viewed as the environment interacting with the exoskeleton in a human–exoskeleton system, a similar impedance relationship exists between the HEI torques $T_{\mathrm{int}}$ and the deviation $\delta q_{r}$ between the actual and reference joint angles.
(4)$$\delta q_{r}=\frac{1}{K_{P}+K_{D}s}T_{\mathrm{int}}$$

By collecting the HEI torque $T_{\mathrm{int}}$, the desired reference joint angle $q_{\mathrm{ref},\mathrm{new}}$ of the patient can be approximated using this relationship, as shown in the following equation.
(5)$$q_{\mathrm{ref},\mathrm{new}}=q_{\mathrm{ref},\mathrm{old}}+w\cdot \delta q_{r}$$
where $q_{\mathrm{ref},\mathrm{new}}$ represents the expected joint reference angle and $q_{\mathrm{ref},\mathrm{old}}$ is nominal reference joint angle, which is the angle value of the $\varphi_{\mathrm{nom}}$ at the current time. w is the moderator, used to change the degree of adaptation of the reference trajectory.

For each sampling moment in a complete gait cycle, the reference angle value expected by the patient at that moment can be calculated by Equation (5). Since human walking is a continuous process, the change in the reference angle at adjacent moments is very small. If the discrete reference angle points calculated at each moment are used as input for gait control, it is easy to produce sudden changes in the reference angle between adjacent moments, resulting in the exoskeleton’s joint motor generating a huge force at one moment, causing secondary injuries to the patient. To avoid this situation, the desired reference angle points calculated at all sampling moments are curve fitted within a complete gait cycle in this paper. As a result, a continuous desired reference gait trajectory curve that corresponds to the current user’s intention can be obtained.

After parameterizing the gait trajectory, it is possible to generate a new gait trajectory by simply changing the trajectory parameters *a* and *c*. As a result, the preceding curve-fitting problem can be recast as an optimization problem. By minimizing the cost function *J*, it is possible to obtain an optimal set of trajectory parameters that generates the fitted desired reference trajectory. The designed cost function is shown as follows.
(6)$$J\left(a_{i},c_{i}\right)=\sum_{k=1}^{2000}{\left\|q_{\mathrm{ref},\mathrm{new}(k)}-\varphi_{i,\mathrm{new}(k)}\right\|}_{2}^{2}$$
where the nominal reference trajectory is divided into 2000 track points with uniform time intervals in a complete gait cycle. k represents the kth sampling moment within a complete gait cycle, i=1 represents the hip joint, i=2 represents the knee joint, $q_{\mathrm{ref},\mathrm{new}(k)}$ is the expected joint reference angle $q_{\mathrm{ref},\mathrm{new}}$ calculated at the kth sampling time calculated by (5), and $\varphi_{i,\mathrm{new}(k)}$ is the value of the reference trajectory curve to be fitted at the kth sampling time, which contains the trajectory parameters *a* and *c*.

In order to ensure that the fitted reference gait trajectory meets the requirements of normal human gait, the following restrictions are placed on the trajectory parameters:(7)$$\begin{array}{l} 0.8\leq a_{i}\leq 1.2\\ 0.8\leq c_{i}\leq 1.2 \end{array}$$

Equations (6) and (7) represent the objective function to be optimized and the constraints, respectively. The random direction method is used in this paper to solve the optimization problem with constraints. Because the optimized reference gait trajectory is in TV space, the motion of the unilateral lower limb can be expressed only by combining the hip and knee reference trajectory curves simultaneously. To circumvent this issue, the two time-varying reference gait trajectories generated by optimization are combined to generate an NTV phase space reference gait trajectory that can contain information about both hip and knee joints [25]. The following is the specific procedure.

Define the angle vector in the NTV phase space as:(8)$$\vec{q}=(q_{h},q_{k})$$
where $q_{h}$ and $q_{k}$ represent the angles of the unilateral lower-limb hip and knee joints, respectively.

As illustrated in Figure 1, the hip reference gait trajectory and the knee reference gait trajectory in TV space have the same period, so the reference trajectory points of both the hip and knee joints at any moment are in one-to-one correspondence; therefore, the reference trajectory points at each moment in TV space can always correspond to generate an NTV phase space joint angle vector:(9)$${\vec{q}}_{t}=(\varphi_{1,\mathrm{new}}(t),\varphi_{2,\mathrm{new}}(t))$$

The curve enclosed by all the phase space angle vectors is referred to as the NTV phase space reference gait trajectory, as described in Figure 2, and it is also the adaptive trajectory generator’s final output. Additionally, this trajectory can be adaptively updated at intervals of one complete gait cycle based on the patient’s intention, laying the groundwork for subsequent balance control.

### 2.2. Inner Loop of the Controller

The outer loop of the controller designed in this paper is responsible for trajectory generation, whereas the inner loop acts as a self-coordinated balance controller based on the outer loop’s NTV phase space reference gait trajectory, which can simultaneously account for the suppleness and tracking of HEI during training while providing balance guiding to the patient. The following describes the specific design process.

The NTV phase space reference gait trajectory obtained from the outer loop is denoted by $Q_{c}$, and the patient’s motion state at any point can be mapped to an NTV phase space joint angle vector $\vec{q}=(q_{h},q_{k})$, referred to as the configuration, where $q_{h}$ and $q_{k}$ denote the unilateral lower-limb hip and knee joints’ angles, respectively. The derivatives of the configuration with respect to time can be expressed as:(10)$$\vec{\omega }=\frac{d\vec{q}}{dt}$$

Assume the current time is $t_{0}$. In TV space, for example, the hip joint is controlled using the current value of the reference trajectory gait curve as the desired hip angle $\varphi_{\mathrm{hip}}(t_{0})$, regardless of the patient’s current hip angle $q_{\mathrm{hip}}(t_{0})$, even if they are far apart. This frequently results in large deviations in the controller’s input, resulting joint torque to exceed patient tolerance. Unlike TV space, NTV phase space is not affected by this issue. In this paper, the reference bit shape corresponding to the current configuration is defined as the point on the NTV phase space reference gait curve where the joint angle ${\vec{q}}_{d}$ at any given moment is closest to the current configuration $\vec{q}$ (in the L2 norm sense). We define the configuration error $\vec{e}$ as:(11)$$\vec{e}={\vec{q}}_{d}-\vec{q}$$

Based on the concept of vector synthesis, the direction of the configuration error $\vec{e}$ is defined as the control normal $\vec{n}$ of the current configuration $\vec{q}$, which is expressed as:(12)$$\vec{n}=\frac{\vec{e}}{\left|\vec{e}\right|}$$

Defining the clockwise direction as the positive direction $\vec{r}$ of the reference gait trajectory in NTV phase space, the tangential direction $\vec{t}$ is always biased towards the positive direction of the reference trajectory. Meanwhile, the normal direction $\vec{n}$ and the tangential direction $\vec{t}$ are mutually perpendicular, so:(13)$$\begin{array}{l} \vec{r}\cdot \vec{t}>0\\ \vec{n}\cdot \vec{t}=0 \end{array}$$

By calculating the length of the configuration error vector $\left|\vec{e}\right|$ and judging the distance between the current patient’s configuration $\vec{q}$ in NTV phase space and the reference configuration ${\vec{q}}_{d}$ corresponding to the NTV phase space reference curve, the effect ratio of the normal and tangential directions can be self-coordinated. Then, the reference joint velocity vector ω is generated by vector synthesis of tangential and normal velocities, and the specific calculation process is shown in the following equation:(14)$${\vec{\omega }}_{\mathrm{ref}}=\left\{\begin{array}{ll} K_{v}(\left|\vec{e}\right|\vec{n}+\frac{\lambda }{\left|\vec{e}\right|}\vec{t}) & \mathrm{for}\left|\vec{e}\right|>0\\ K_{v}\vec{t} & \mathrm{for}\left|\vec{e}\right|=0 \end{array}\right.$$
where ${\vec{\omega }}_{\mathrm{ref}}$ denotes the reference velocity vector, its value in the x direction indicates the magnitude of the hip reference velocity, and its value in the y direction indicates the magnitude of the knee reference velocity. $K_{v}$ is a scalar to determine the magnitude of the reference velocity. In order to ensure that the magnitude of the reference joint velocity vector ${\vec{\omega }}_{\mathrm{ref}}$ is only affected by $K_{v}$, after calculating $\left(\left|\vec{e}\right|\vec{n}+\frac{\lambda }{\left|\vec{e}\right|}\vec{t}\right)$, it needs to be normalized so that $\left(\left|\vec{e}\right|\vec{n}+\frac{\lambda }{\left|\vec{e}\right|}\vec{t}\right)$, like $\vec{t}$, is a unit direction vector, providing only directional information. λ is the velocity direction adjustment factor. By adjusting the relationship between the magnitudes of $\frac{\lambda }{\left|\vec{e}\right|}$ and $\left|\vec{e}\right|$, the magnitudes of the vectors in the normal and tangential directions can be changed so that the reference velocity vectors in different directions can be obtained by vector synthesis.

Each vector mentioned in Equation (14) is depicted in Figure 3. The normal direction is correction, which indicates the direction of action to correct the current motion of the patient. Movement in this direction could be the quickest way to the desired reference trajectory, allowing the patient’s non-physiological gait to be corrected as quickly as possible and the patient to reach the physiological reference gait as soon as possible. The tangential direction is encouragement, indicating the direction of action that encourages the patient’s current movement. Movement in this direction maintains the physiology of the patient’s gait. As the geometric relationship indicates, the tangential direction $\vec{t}$ is identical to the positive direction $\vec{r}$ of the reference gait trajectory at ${\vec{q}}_{d}$; it is advantageous to move in that direction. Even if the length of the configuration error vector $\left|\vec{e}\right|$ is not equal to zero at that moment, the same gait trend as the reference trajectory can be maintained as long as the movement is tangential.

## 3. Simulation Model and Result

In this part, in order to verify the effectiveness of the SCVV controller proposed in this paper, a simulation model of the LLRER was built using the MATLAB/Simulink simulation software, and the specific implementation of the simulation is as follows.

### 3.1. Simulation Model

In view of the fact that the human lower limb is a multi-degree-of-freedom, nonlinear structure, the mathematical model is very complex. In order to facilitate the analysis, modeling, and control, combined with human characteristics, a simple two-link-based LLRER model was established.

As shown in Figure 4, the left side is the sagittal structure of the exoskeleton model, and the right side is the frontal structure. The model mainly consists of two joint links at the hip and knee; $l_{1}$ and $l_{2}$ represent the lengths of the thigh link and the calf link, respectively. The angle between the thigh rod and the vertical direction is $q_{1}$, and it turns clockwise to be positive and counterclockwise to be negative. The angle between the extension of the thigh rod and the calf rod is recorded as the knee joint angle $q_{2}$; its clockwise rotation around the thigh rod is recorded as positive and, counterclockwise rotation as negative.

The robot dynamics model is used to describe the dynamic relationship between the motion state (position, velocity, acceleration) and the drive torque of each joint of the robot. The two-degree-of-freedom robot dynamics are:(15)$$T=M(q)\ddot{q}+C(q,\dot{q})\dot{q}+G(q)+D(q)+T_{\mathrm{int}}$$
where $M(q)\in R^{4\times 4}$ is a symmetric positive definite inertia matrix; $C(q,\dot{q})\in R^{4}$ is the Coriolis force and centrifugal force matrix; $G(q)\in R^{4}$ is the gravity vector; $D(q)\in R^{4}$ is the friction torque matrix; and $T_{\mathrm{int}}\in R^{2\times 1}$ is the HEI torque matrix between the user and the LLRER. $q\in R^{2\times 1}$, $\dot{q}\in R^{2\times 1}$, and $\ddot{q}\in R^{2\times 1}$ are the position, velocity, and acceleration of the robot in the joint space, respectively, obtained by the robot joint encoder.

Combined with the brushless DC motor model, the electromechanical coupling model of the robot system can be further obtained.
(16)$$J\frac{d\omega }{dt}=T_{e}-(M(q)\ddot{q}+C(q,\dot{q})\dot{q}+G(q)+D(q)+T_{\mathrm{int}})$$
where $J\in R^{2\times 2}$ is the rotational inertia matrix, $\omega \in R^{2\times 1}$ is the motor angular velocity matrix, and $T_{e}\in R^{2\times 1}$ is the electromagnetic torque matrix.

### 3.2. Control Framework and Simulation Setup

The overall control framework of the LLRER is shown in Figure 5 and consists of three main parts: the controller outer loop structure, the controller inner loop structure, and the human–machine system model.

The outer loop generates an NTV phase space reference gait trajectory by collecting and optimizing the HEI torque information and inputting it into the inner loop of the controller. Based on this trajectory, the inner loop of the controller can generate the self-coordinated reference joint velocity vector. The underlying control drives the joint motor, which in turn drives the human–machine system to perform the motion.

In order to obtain more reliable results in the simulation, the LLRER model built in the simulation was completely parameterized according to the real human body and the data of the LLRER. The specific parameters are shown in Table 1.

The motor model used in Simulink was a brushless DC motor, and its parameters were consistent with the motor parameters selected in the experiment. The specific settings were: rated voltage was 24 V, rated speed was 7220 rpm, rated torque (maximum continuous torque) was 33.8 mN·m, phase-to-phase resistance was 3.65 Ω, phase-to-phase inductance was 0.31 mH, torque constant was 24.3 mN·m/A, rotor moment of inertia was 11 g·cm^2^, the number of motor pole pairs was 1 pair, and the number of phases was 3. Considering the design of the mechanical transmission structure of the LLRER, the reduction ratio was set to 30.

Since the HEI torque between the patient and the exoskeleton cannot be measured in the simulation, the definition of trajectory parameterization was combined with a target trajectory curve to replace the user’s intention in this paper, and the human–robot HEI torque was represented by calculating the difference between the target trajectory curve and the actual trajectory curve. The specific calculation process is shown as follows:(17)$$\varphi_{i,\mathrm{target}}(t)=a_{i,\mathrm{target}}\varphi_{i,\mathrm{nom}}\left(\frac{t}{c_{i,\mathrm{target}}}\right)\;i=1,2$$
(18)$$T_{i,\mathrm{int}}=K_{t}(q_{i,\mathrm{target}}-q_{i})$$

Among them, $\varphi_{i,\mathrm{target}}(t)$ is the target trajectory curve used to calculate the human–computer HEI force, and on the basis of the nominal trajectory $\varphi_{i,\mathrm{nom}}$, the specific values are determined by introducing the trajectory parameters $a_{i,\mathrm{target}}$ and $c_{i,\mathrm{target}}$; $T_{i,\mathrm{int}}$ is the HEI torque between the patient and the exoskeleton. $q_{i,\mathrm{target}}$ and $q_{i}$, respectively, represent the value of the target trajectory curve at a certain moment and the actual joint radian value of the patient at that moment. $K_{t}$ is the proportional factor for calculating the HEI torque.

### 3.3. Simulation Results of Adaptive Gait Generation 

In order to verify the effectiveness of the control outer loop, we use Equation (1) to generate the target trajectory curve and the reference trajectory curve. In this section, the trajectory parameters of the target trajectory curve are set as follows: $a_{1,\mathrm{target}}=1.15$, $a_{2,\mathrm{target}}=1.05$, and $c_{1,\mathrm{target}}=c_{2,\mathrm{target}}=0.95$, where $a_{1,\mathrm{target}}$ and $c_{1,\mathrm{target}}$ are the target trajectory curve parameters of the hip joint, and $a_{2,\mathrm{target}}$ and $c_{2,\mathrm{target}}$ are the target trajectory curve parameters of the knee joint. The initial trajectory parameters of the reference trajectory are $a_{1}=1$, $a_{2}=1$, and $c_{1}=c_{2}=1$, where $a_{1}$ and $c_{1}$ are the reference trajectory curve parameters of the hip joint, and $a_{2}$ and $c_{2}$ are the reference trajectory curve parameters of the knee joint. The impedance relationship parameters in the control implementation process were set to $B_{1}=1$, $K_{1}=44$, $B_{2}=1$, and $K_{2}=32$, where $B_{1}$ and $K_{1}$ are the impedance relationship parameters of the hip joint, and $B_{2}$ and $K_{2}$ are the impedance relationship parameters of the knee joint.

Based on the above parameter settings, the outer loop adaptive trajectory generation algorithm was simulated and verified for 10 s, as shown in Figure 6.

There is a significant deviation between the target trajectory curve representing the patient’s intention and the reference trajectory curve during the initial gait cycle. Beginning with the second gait cycle, the controller’s adaptive adjustment brings the reference gait trajectory closer to the target. In the third cycle, the reference trajectory nearly perfectly matches the target reference trajectory, indicating that the updated reference trajectory meets the patient’s expectations.

The trajectory parameters in Figure 7 correspond to the trajectory curves in Figure 6, and can more intuitively reflect the adaptive trajectory generation algorithm’s rapidity and effectiveness. The three subgraphs illustrate the changes in the trajectory parameters $a_{1}$, $a_{2}$, and c. The red dotted lines represent the target values for the corresponding parameters, i.e., the target for the adaptive trajectory parameter update, whereas the solid black lines represent the current reference gait trajectory’s actual parameter values. Because the trajectory parameters are updated on a periodic basis, the value of the reference gait trajectory remains unchanged during the cycle and is updated only after the next cycle. After ten seconds of simulation, the closeness between the trajectory parameters $a_{1}$, $a_{2}$, and c was determined, and the target value was greater than 95%, confirming the accuracy and reliability of the trajectory adaptive outer loop.

Figure 8 shows the change diagram of the HEI torque of the hip and knee joints during the reference trajectory adaptation process. In the initial gait cycle, HEI torque was large, and the fluctuation was more obvious. With the adaptive adjustment of the controller, the HEI torque kept decreasing, gradually approaching 0, which verifies the effectiveness of the trajectory adaptive generator.

### 3.4. Simulation Results of the Self-Coordination Balance Guiding

As shown in Equation (14), there are three parameters, $K_{v}$, λ, and $\left|\vec{e}\right|$, that affect the controller’s self-coordination performance. Among them, $K_{v}$ and λ are control parameters. In order to avoid the instability that may be caused by parameter adaptation, control parameters $K_{v}$ and λ are generally set before the control and remain unchanged during the control process. The self-coordination ability of the controller in the control process depends more on the ability to change the configuration error $\left|\vec{e}\right|$ in real time.

The effect of the configuration error $\left|\vec{e}\right|$ is first evaluated. We set parameter $K_{v}$ to 1.3 and parameter λ to 0.02 by changing the actual value of the joint angle vector in order to study the self-coordination ability of the configuration error $\left|\vec{e}\right|$ on “encouragement” and “correction”, as well as the effect on the final output reference velocity vector.

The proportion of the encouragement and correction direction is determined by the size of $\frac{\lambda }{\left|\vec{e}\right|}$ and $\left|\vec{e}\right|$, as shown in Figure 9. When control parameters $K_{v}$ and λ are fixed, the proportion of encouragement gradually increases as the configuration error $\left|\vec{e}\right|$ increases, and the correction effect grows. With increasing the configuration error $\left|\vec{e}\right|$, the proportion decreases. On the contrary, when the configuration error $\left|\vec{e}\right|$ is large, it indicates that the patient is unreasonable and needs to be adjusted quickly. Otherwise, it affects the balance, so the proportion of the corrective effect is high at this time; furthermore, when the configuration error $\left|\vec{e}\right|$ is small, the proportion of the encouraging effect is high, indicating that this patient is reasonable and needs to be adjusted soon. When the patient’s gait is more physiological, the controller encourages the patient to continue walking in the direction provided by the reference trajectory, and, on that basis, it performs a small gait correction to ensure the patient’s gait coordination and balance.

The configuration error $\left|\vec{e}\right|$ changes with the change in the patient’s current joint angle vector, which directly affects the ratio of encouraging and correcting direction effects, but what ultimately changes is the output of the controller—the reference velocity vector. Figure 10 describes the influence of the configuration error $\left|\vec{e}\right|$ on the controller output and introduces the reference velocity vector directions at different points, which is the same as the above analysis, which also reflects the adjustment of the self-coordination performance derived from the control strategy designed based on the configuration error $\left|\vec{e}\right|$.

In order to study the specific effects of control parameters $K_{v}$ and λ on the output, this section simulates the process of a patient from standing to walking a complete gait cycle under different $K_{v}$ and λ conditions and records the time spent by the patient from standing to entering the NTV phase space reference domain. The so-called NTV phase space reference domain, i.e., the area region of ±2 degrees around the NTV phase space reference trajectory, is shown in blue in Figure 11, and the patient is considered to have reached a relatively physiological gait when the patient’s current configuration enters the NTV phase space reference domain.

Figure 12 depicts the time taken by the patient from standing to moving into the NTV phase space reference domain under different control parameters $K_{v}$ and λ conditions. Combined with Equation (14), when $K_{v}$ is constant, and λ is larger, the synthesized output reference velocity direction is more tangential, and the controller provides more encouragement, not forcing the patient to stay close to the reference trajectory, so it takes more time for the patient to enter the NTV phase space reference domain. In contrast, when λ is small, the synthesized output reference velocity direction is more inclined to the normal direction, and the controller provides more correction along the “normal direction” to pull the patient closer to the reference trajectory, so that it enters the NTV phase space reference domain more quickly and achieves a physiological gait.

The effect of the control parameter $K_{v}$ on the training outcome is not monotonic, as opposed to the effect of the control parameter λ, so when discussing parameter $K_{v}$, two cases must be divided. When λ is larger, as shown in the right part of Figure 12, the time for the patient to enter the reference domain increases as $K_{v}$ increases. Because an increase in $K_{v}$ causes the patient to move faster in the off-tangential velocity direction, the patient maintains the current gait longer and enters the NTV phase space reference domain later; the opposite case, as shown in the left part of Figure 12, is that when λ is small, the corrective effect of the controller is stronger because of the increasing value of $K_{v}$ at this time, so the patient approaches the NTV phase space reference domain at a faster speed and thus achieves the desired result.

Tracking performance is a critical performance indicator for the controller of the LLRER. Comparative simulations of three control methods, namely, position control, impedance control, and SCVV control, were conducted using the same reference trajectory. The average error values between the patient’s actual joint angle vector and the patient’s reference joint angle vector are recorded throughout a gait cycle, and the results are shown in Figure 13.

As shown in Figure 13, in a complete gait cycle, the average error of position control is the smallest, indicating that its overall tracking effect is the best; the average error of impedance control is the largest because the purpose of this control is to maximize compliance, so part of the tracking performance is sacrificed; and the average error of SCVV control over a complete gait cycle is between that of position control and impedance control. It shows that although SCVV control does not aim to track trajectory, it has good tracking performance.

Meanwhile, the upper error limit of the error bars in Figure 13 presents the average error of the first half cycle, and the lower error limit represents the average error of the second half cycle. It can be seen that under SCVV control, the average error in the first half cycle is significantly larger than that in the second half cycle, indicating that the patient is gradually approaching the NTV phase space reference trajectory and the gait has improved.

Under the same conditions as in the tracking performance simulation, the HEI torque between the patient’s hip, knee joint, and exoskeleton under a complete gait cycle was collected, and the compliance of the three control methods was compared.

The results are shown in Figure 14a,b, respectively. It can be clearly seen from the figure that in a complete gait cycle, under the condition of position control, the fluctuation and amplitude of the HEI torque of the hip and knee joints are the largest, while the smallest is the impedance control. The HEI torque generated by SCVV control is between position and impedance control. Table 2 shows the specific statistics for Figure 14.

As shown in Table 2, the average and standard deviation of the HEI torque of the position control on the hip and knee joints are particularly high when compared to the SCVV control and the impedance control, indicating that the fluctuation is large, and the maximum value can reach 23 Nm, which can easily cause injury to the patient. The impedance control has the best compliance performance, and the data performance of the self-coordinated control is not significantly different from that of the impedance control, indicating that it, too, has good compliance.

It is also a comparative simulation. In order to test the balance-guiding performance of the SCVV controller of the LLRER, the following simulation design was carried out: The patient started normally, a disturbance force of 200 N was added to the patient’s hip protrusion between 0.15 s and 0.2 s, and different electromagnetic torque variation curves were obtained under each of the three control strategies.

Figure 15 shows the variation curve of the electromagnetic torque of the hip joint motor before and after a sudden disturbance. Among them, the position control and impedance control have obvious electromagnetic torque mutations at 0.15 s and 0.2 s, which indicates that under these two controls, the joint motor will output a large force to resist external disturbance in a moment, which usually leads to the patient’s imbalance and even falling down. However, when the SCVV controller suddenly encounters a disturbance, the electromagnetic torque of its joint motor changes relatively gently. This is because the controller does not force the patient to track and also allows a configuration error between the patient and the reference trajectory in NTV phase space. At the same time, the configuration error is gradually reduced by the SCVV of the encouraged and corrected directions. This allows the patient to adjust the hip and knee joint angles in time to maintain body balance through their own judgment of balance in an instant when they encounter a disturbance in balance. The patient will not be over-torqued by the controller in an instant due to deviation from the reference trajectory, resulting in imbalance.

## 4. Units Experimental Implementation and Evaluation

### 4.1. Exoskeleton Hardware and Experimental Setup

The mechanical structure of the exoskeleton hardware platform is shown in Figure 16, which is composed of four parts: computer terminal, motion control system, motion and information acquisition system, and exoskeleton components. The algorithm instruction is sent out by the computer terminal and transmitted to the control system through EtherCAT Bus, and the motion of the joint motor is controlled by the driver. The joint motor is used to drive the external exoskeleton structure, collect the exoskeleton joint state information through the acquisition system, and feed it back to the motion control system and PC terminal for further processing.

The joint motor is connected to the computer end through a complete set of driving equipment, and the real-time joint angle of the patient is obtained through the absolute value encoder at the motor end.

In order to ensure the safety of the experiment, experiments were conducted on healthy subjects using the LLRER exoskeleton device. Unless otherwise specified, all experimental subjects consisted of three adult males with an approximate height of 175 cm and a weight of 60 kg. The experimental procedure involving human subjects in this article has been approved by the Institutional Ethics Committee of the First Hospital of Qinhuangdao, China. During the experiment, the subjects only wore a single right-leg LLRER, and the left leg was freely controlled by the human body. At the same time, the motor control gain was set within a reasonable and safe range, and the experimenter’s consent was obtained before each experiment was performed.

### 4.2. Self-Coordinated Experimental Evaluation

Based on the simulation conclusion, the self-coordination ability of the SCVV controller is mainly regulated by the control parameter λ. Therefore, based on the exoskeleton device, experiments were designed to further verify the influence of parameter λ on the self-coordination ability of the controller. The specific experimental design is as follows:Experimental Preparation: Prepare and lay out a spacious and flat pathway. Participants wear and familiarize themselves with the LLRER exoskeleton device.Experimental Procedure: Three participants sequentially wear the LLRER exoskeleton device and repeat walking on the predetermined pathway. Each participant performs three sets of walks, with a duration of 20 s per set.Condition Control: Among the sets performed by each participant, the only differing factor is the control parameter, while all other parameters and conditions remain consistent. The values of the parameter for the three sets of experiments are 0.02, 0.2, and 2, respectively.Participant Awareness: Participants are not informed that the experiment’s outcomes are influenced by their own actions, nor are they informed of any changes in the control parameters.Data Collection: Collect position trajectories of the participants’ knee and hip joints, as well as motor speed profiles.

The experimental results are shown in Figure 17 and Figure 18.

Figure 17 shows the deviation between the experimenter’s actual walking trajectory and the reference trajectory in NTV phase space when λ varies, and the specific error value is shown in Figure 17d. It can be seen that when λ is smaller, the actual walking trajectory is closer to the reference trajectory. At this point, the controller primarily provides encouraging support, allowing the patient to gradually approach the reference trajectory without changing their current gait. If λ is large, the actual walking trajectory is far from the reference trajectory. The controller primarily corrects the user’s current gait at this point to bring it closer to the physiological reference trajectory. This reflects the effect of λ on the actual control process and corresponds to the simulation results.

The changes in the rotational speed of the joint motor are depicted in Figure 18, where Figure 18a depicts the rotational speed of the hip motor and Figure 18b depicts the rotational speed of the knee motor. Because λ only modifies the direction of the output reference speed, it has no effect on the reference speed. As a result, it can be seen in the joint motor that under the three conditions of λ, whether it is the hip motor or the knee motor, there is no significant difference in the overall speed and trend; only a difference in phase. When λ is small, the experimenter walks closer to the reference trajectory faster, and the joint motor’s speed completes one cycle of operation faster and enters the next cycle faster.

### 4.3. Balance Guiding Experimental Evaluation

The following comparative experiments were carried out to validate the benefits of the proposed indirect balancing strategy in the balance guiding of the LLRER. First, the experimenter was asked to walk normally while wearing the LLRER, and then the experimenter was asked to suddenly increase the angle of the hip joint by raising the leg, which was equivalent to simulating the experimenter’s emergency response in the face of a balance disturbance. Position control, impedance control, and SCVV control, as proposed in this paper, are used to control the LLRER under such conditions. The experimental results are shown in Figure 19.

Figure 19a depicts the change in the hip motor current before and after the experimenter suddenly increases the hip joint angle, and Figure 19b shows the change in the hip motor speed before and after the experimenter suddenly increases the hip joint angle. It can be seen from the figure that the experimenter suddenly increased the angle of the hip joint at about 1.4 s. The changes in the current of the hip joint motor under the three control methods are not very obvious, and there is almost no difference, but in the speed of the hip joint motor, there is a clear difference in the control. 

Among them, the motor speed of the position control method and the impedance control method shake significantly. For the position control, at a certain moment, the experimenter is disturbed, which causes the actual trajectory to deviate from the reference trajectory, and the motor needs to apply additional force to combat this bias.

For impedance control, once there is an interference force, the controller adjusts the reference trajectory so that the experimenter moves according to the new reference trajectory. In order to respond to this change, the motor produces speed jitter in response. For SCVV control, the existence of instantaneous configuration deviation is allowed, and the experimenter is not forced to follow a fixed trajectory. Therefore, in the case of balance interference, the experimenter can adjust the balance by moving the joint position without being hindered by the controller. Meanwhile, the controller coordinates the movement between the hip and knee joints according to the actual joint angle of the current joint so as to ensure the balance.

### 4.4. Step Size and Pace Experiment

Stride length and walking speed also have a significant impact on human gait. An experiment was designed to investigate the regulatory performance of the SCVV controller on stride length and walking speed. The experimental description is as follows:Experimental Preparation: Prepare and lay out a spacious and flat walkway with a length of 25 m.Experimental Procedure: Three participants sequentially wear the LLRER exoskeleton device and walk a distance of 20 m on the designated walkway. Each participant performs five sets of walks.Condition Control: Within each set of experiments, participants are instructed to maintain the required speed as closely as possible. Among different sets of experiments for the same participant, only the walking speed varies. The walking speeds for the five sets of experiments are: 0.6 m/s, 0.7 m/s, 0.8 m/s, 0.9 m/s, and 1 m/s, respectively.Data Collection: Measure the stride length of each step taken by the participants and record the walking speeds of the participants.

The experimental results are shown in Figure 20. There are five groups of experiments, in which the bar graph represents the average step length of walking 20 m under a certain group of pace conditions. The line graph shows the actual average pace of the experimenter in these five groups of experiments. It can be seen that the actual average pace of each group is close to the pace required by the group, indicating that the experiment is more accurate.

Analysis of Figure 20 shows that no matter what pace the experimenter uses to complete the 20 m experiment, the step length remains around 0.58 m. The change in pace indicates that in the process of walking, the gait trajectory of the experimenter changes accordingly. However, at different paces, the experimenter’s step size was almost unchanged, which indicated that the experimenter’s movement results were the same for each step. The above results show that the SCVV controller proposed in this paper only adjusts the joint angle during the movement process and does not affect the final experimental purpose. In other words, the SCVV controller can ensure that the patient achieves the predetermined training goal while also providing the patient with the ability to guide balance.

In order to study the control effect of the SCVV controller on individuals with varying heights and weights, the following experiment was devised. The experimental description is as follows:Subject Information: Four healthy adult males with heights of 1.65 m, 1.7 m, 1.75 m, and 1.8 m, corresponding to body weights of 60 kg, 68 kg, 75 kg, and 80 kg, respectively.Experimental Procedure: The four subjects sequentially wore LLRER exoskeleton devices and walked back and forth on a predetermined 25 m pathway. This constituted one experimental group, and a total of five groups were conducted.Condition Control: Within each group, the subjects were instructed to maintain the required speed, while different groups varied only in walking speed. The control parameters remained the same across all groups. The walking speeds for the five groups were 0.6 m/s, 0.7 m/s, 0.8 m/s, 0.9 m/s, and 1 m/s, respectively.Data Collection: The stride length of each step and the walking speed of the subjects were measured. To minimize the influence of starting and stopping on the experimental results, the first and last two steps near the beginning and end of the pathway were excluded from the statistical analysis.

The experimental results are shown in Figure 21.

Analyzing Figure 21, it can be observed that with the same walking speed, the average stride length increases with the height of the subjects. This indicates that individuals with different heights and weights have a relatively similar range of hip joint rotation during walking. This is because the length of the lower limbs in the human body is positively correlated with height. When walking, subjects lift their hip joints at the same angle, resulting in a longer stride length for taller individuals. This result reflects that the control effect of the SCVV controller is not influenced by the user’s height and weight. The exoskeleton device using this controller can provide the same rehabilitation effect for individuals with different heights and weights. For the same subject, when walking at different speeds, the stride length varies within a reasonable range. The variation in stride length indicates that the controller updates the desired trajectory based on the actual movement trajectory of the lower limbs, thereby providing balance assistance to patients. This finding aligns with the conclusions drawn from the experiment shown in Figure 20, demonstrating that the proposed controller does not sacrifice gait physiology for faster walking speed, while still providing balance guidance to patients.

## 5. Conclusions

This paper describes a controller for an LLRER that solves the balance problems posed by traditional controllers. The controller has a double-loop structure, and the outer loop is responsible for providing a harmonious hip–knee joint reference trajectory based on an NTV phase space. The inner loop encourages the direction and corrects the direction through self-coordination, synthesizes a reasonable reference velocity vector, and provides balanced guiding for the patient, so it is called the SCVV double-loop controller. In both the designed simulations and experiments, the self-coordinated controller was compared with the position control and the impedance control. The results indicate that the SCVV controller can significantly improve balance-guiding capability while maintaining a high level of compliance and tracking performance.

This study still has several aspects that need further improvement. In the future, efforts will be focused on enhancing the control strategy based on the following two points:In the inner loop self-coordinated velocity vector control of the controller, both control parameters need to be determined prior to control and cannot be adapted based on the user’s current condition. Subsequent research will investigate how to achieve adaptive control parameters.The mentioned hierarchical balance-guided control strategy heavily relies on the phase space reference gait trajectory generated by the outer loop. To address this issue, future considerations will involve the use of intelligent optimization algorithms to modify the reference gait trajectory, thereby further ensuring the balancing performance of the controller.

## Figures and Tables

**Figure 1 sensors-23-05311-f001:**
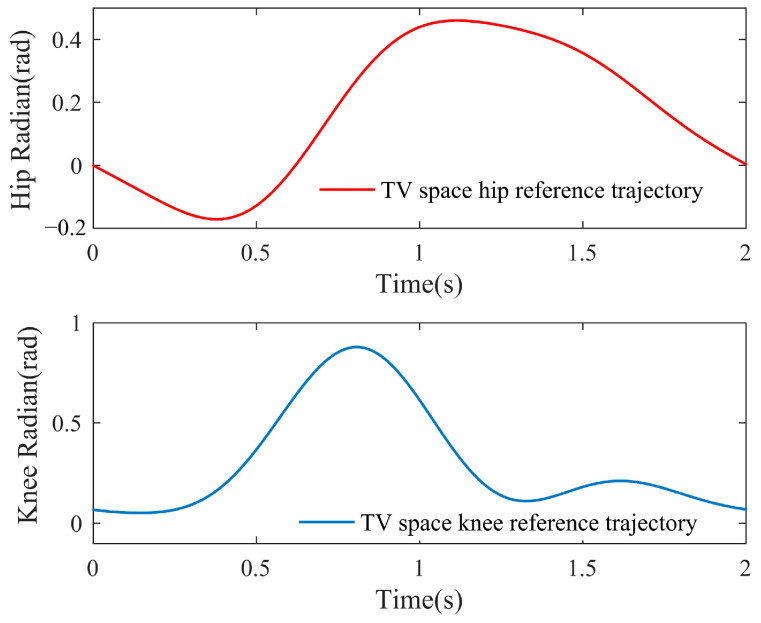
TV space reference trajectory.

**Figure 2 sensors-23-05311-f002:**
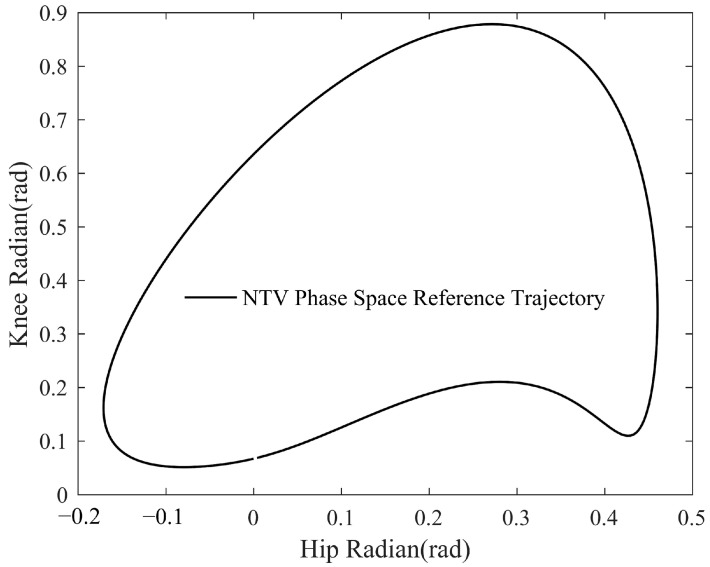
NTV phase space reference trajectory.

**Figure 3 sensors-23-05311-f003:**
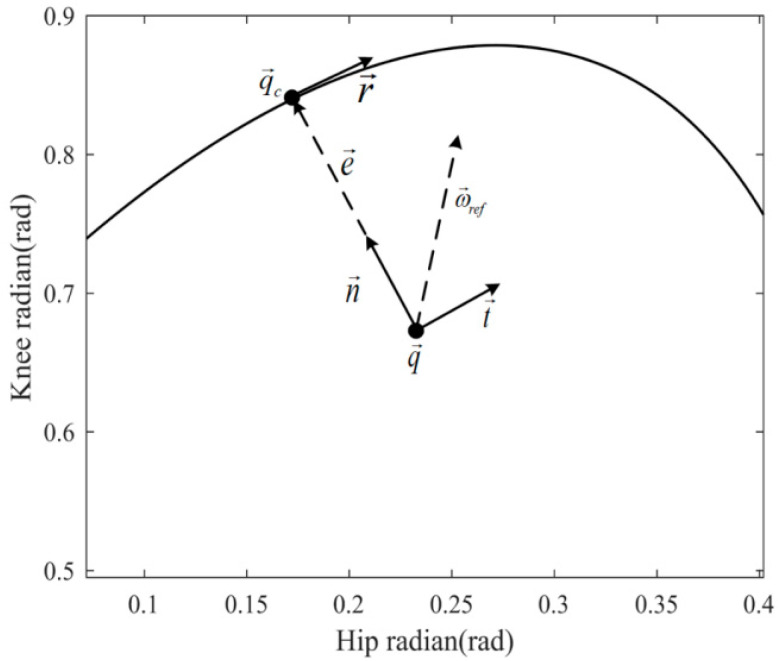
Description of each vector in NTV phase space.

**Figure 4 sensors-23-05311-f004:**
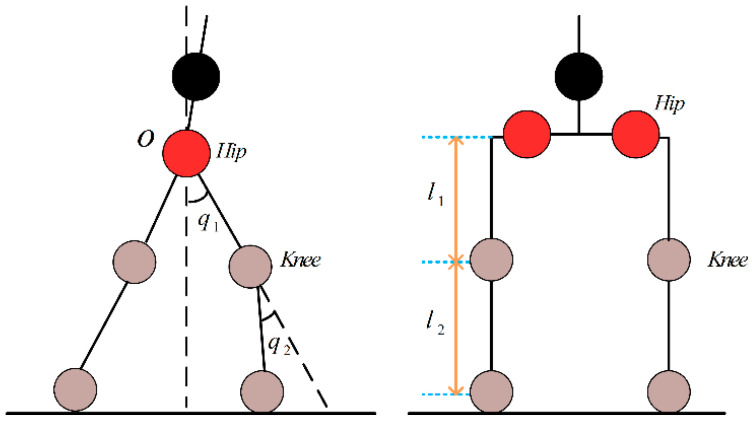
Simple model of unilateral lower limb of LLRER.

**Figure 5 sensors-23-05311-f005:**
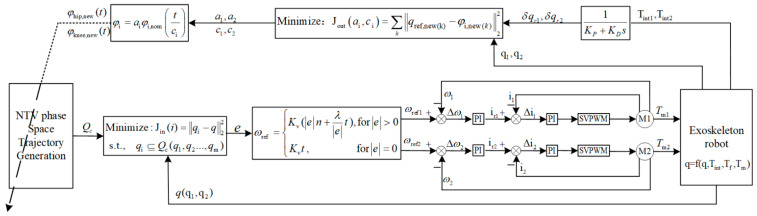
The overall control framework of the LLRER.

**Figure 6 sensors-23-05311-f006:**
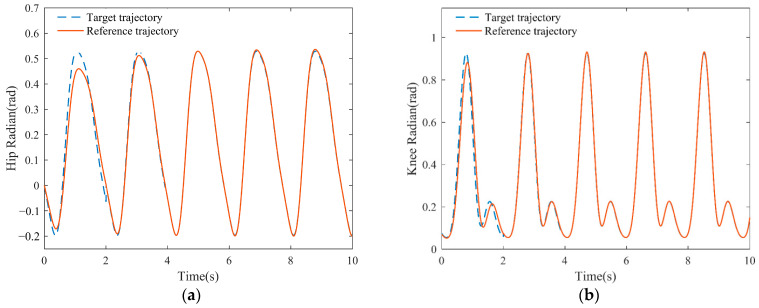
Simulation results of gait trajectory adaption: (**a**) Hip reference gait trajectory adaption; (**b**) Knee reference gait trajectory adaption.

**Figure 7 sensors-23-05311-f007:**
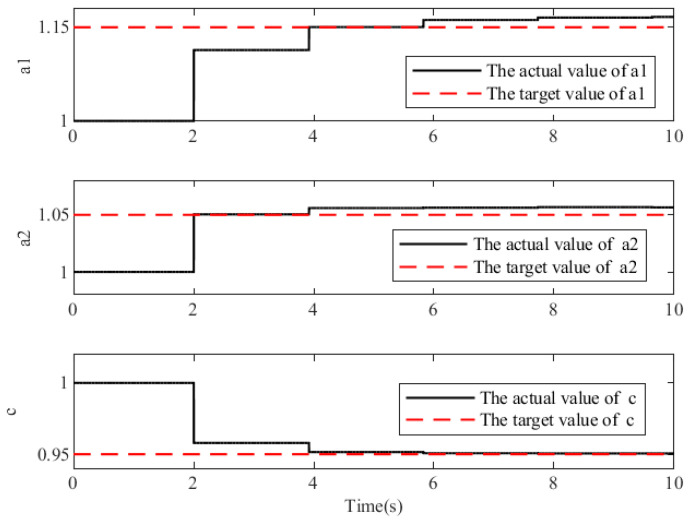
Simulation results of gait trajectory parameters.

**Figure 8 sensors-23-05311-f008:**
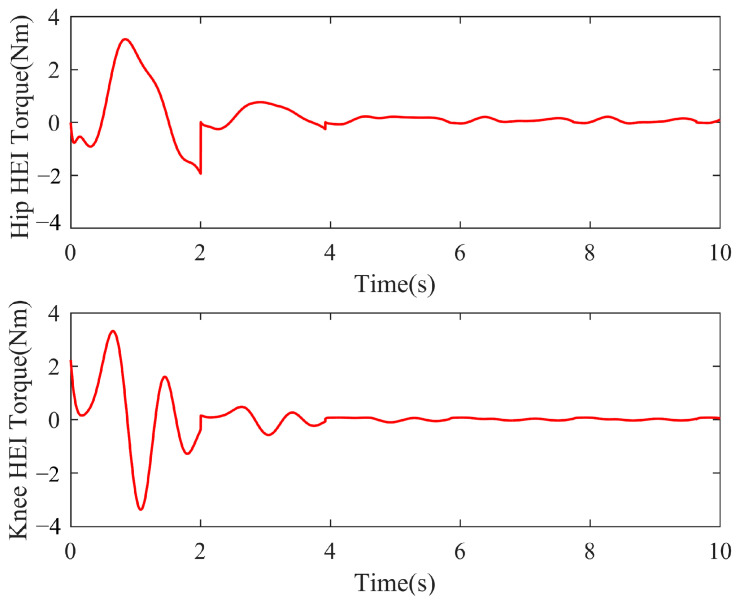
Simulation results of hip and knee HEI torque.

**Figure 9 sensors-23-05311-f009:**
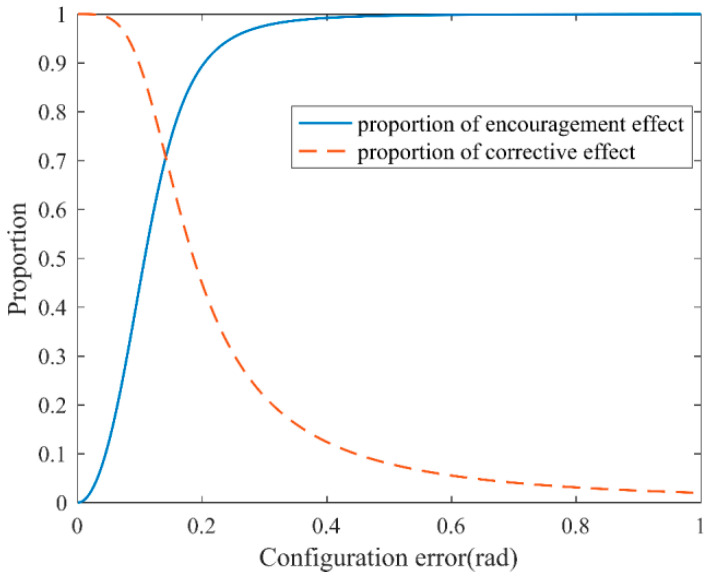
The effect of $\left|\vec{e}\right|$ on self-coordination performance.

**Figure 10 sensors-23-05311-f010:**
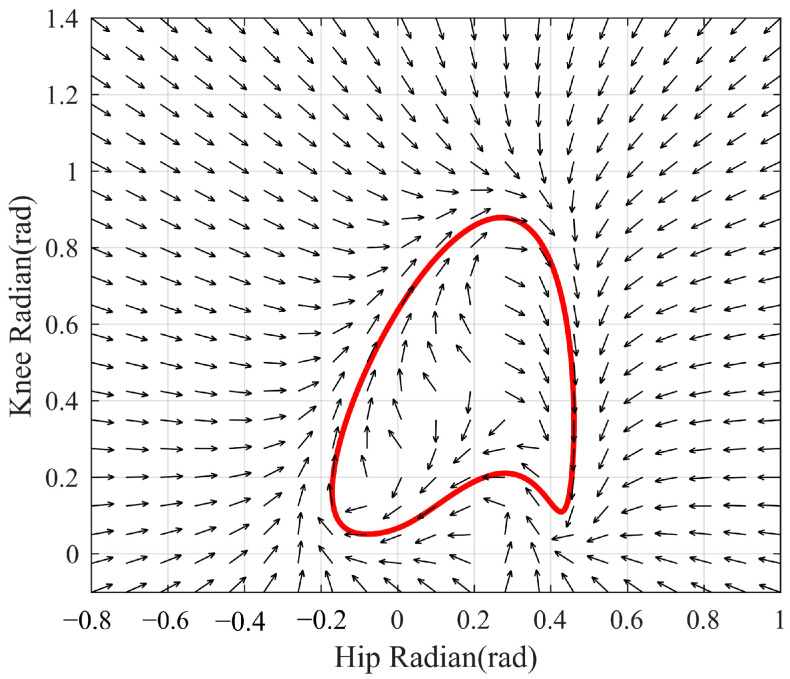
Reference velocity vector direction at different points.

**Figure 11 sensors-23-05311-f011:**
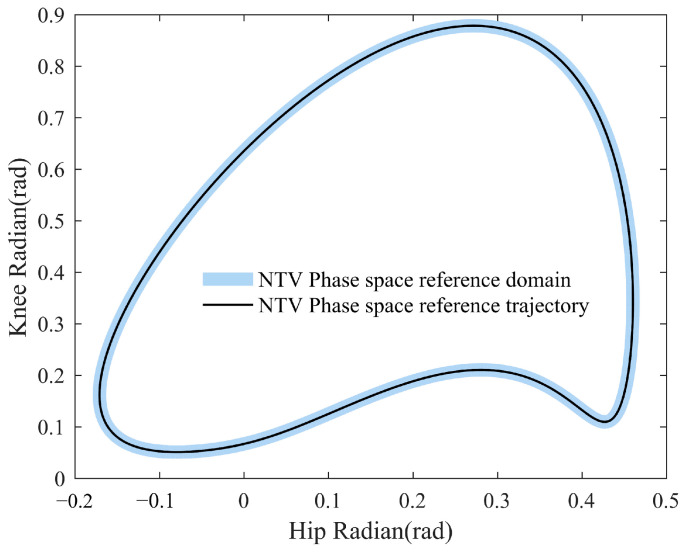
Schematic diagram of NTV phase space reference domain.

**Figure 12 sensors-23-05311-f012:**
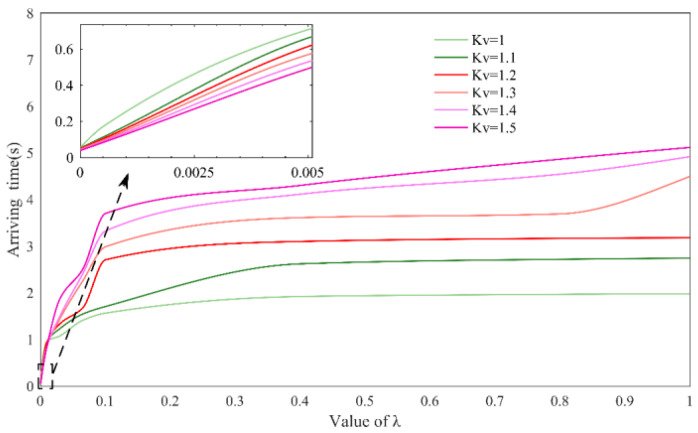
Effect of control parameters $K_{v}$ and λ on gait trajectory.

**Figure 13 sensors-23-05311-f013:**
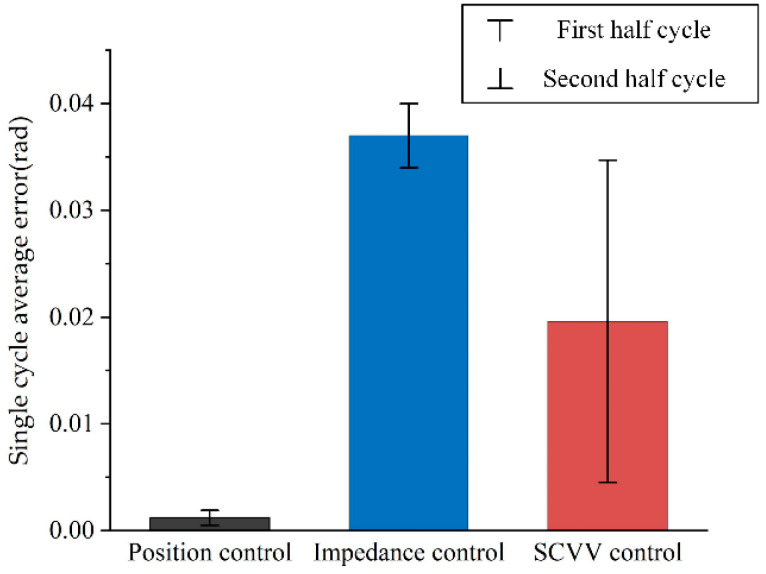
Comparison of individual gait cycle tracking errors.

**Figure 14 sensors-23-05311-f014:**
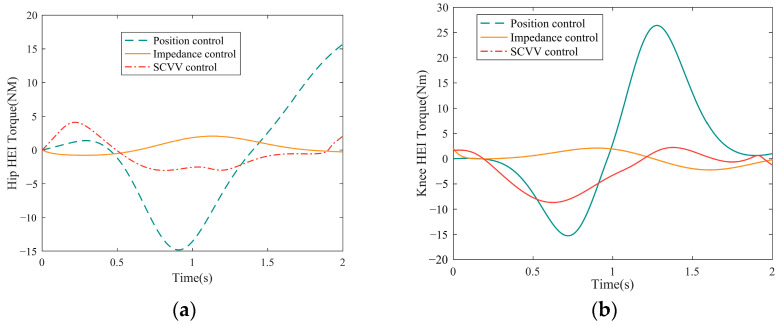
(**a**) Single gait cycle hip HEI torque; (**b**) Single gait cycle knee HEI torque.

**Figure 15 sensors-23-05311-f015:**
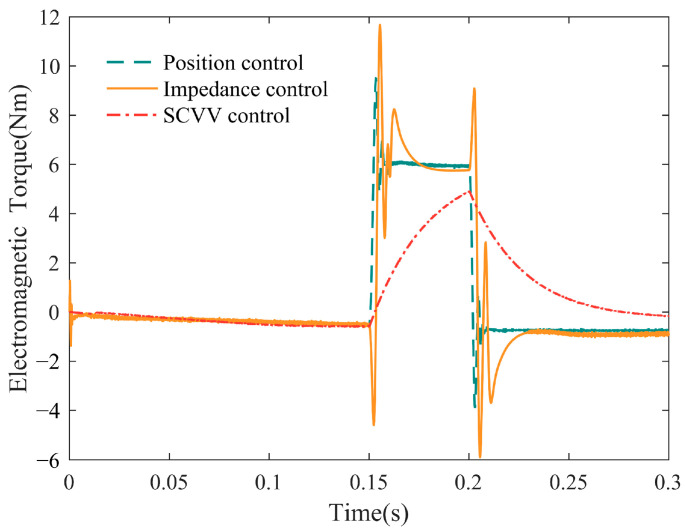
Electromagnetic torque before and after sudden disturbance.

**Figure 16 sensors-23-05311-f016:**
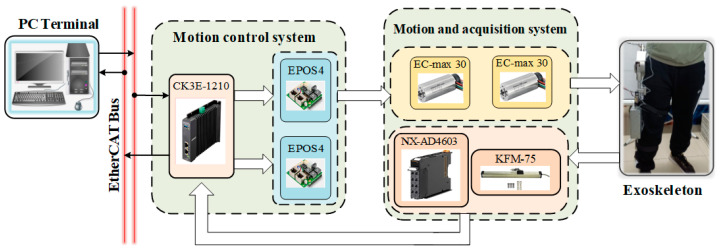
Experimental platform hardware structure.

**Figure 17 sensors-23-05311-f017:**
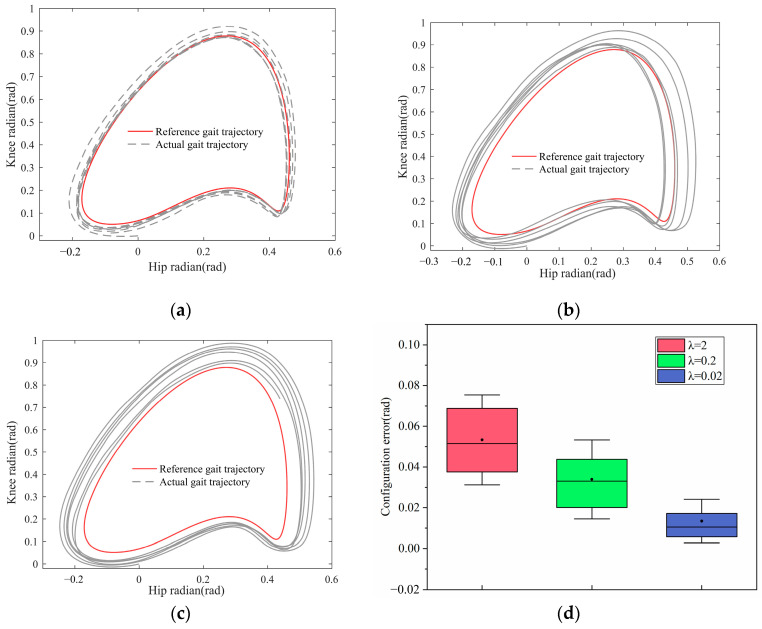
The deviation between the experimenter’s actual walking trajectory and the reference trajectory in NTV phase space when λ varies, the meaning of each part of the figure is as follows: (**a**) Actual gait trajectory when λ=0.02; (**b**) actual gait trajectory when λ=0.2; (**c**) actual gait trajectory when λ=2; and (**d**) multi-cycle walking trajectory error.

**Figure 18 sensors-23-05311-f018:**
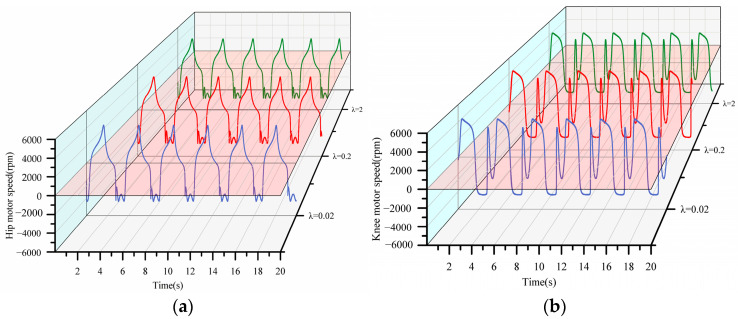
(**a**) Speed of the hip motor; (**b**) speed of the knee motor.

**Figure 19 sensors-23-05311-f019:**
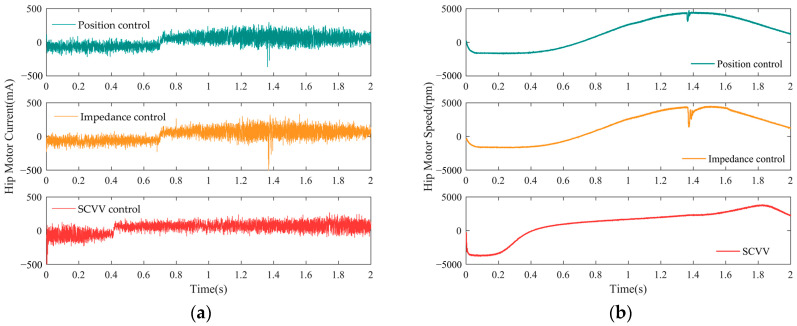
(**a**) Current of the hip motor; (**b**) speed of the hip motor.

**Figure 20 sensors-23-05311-f020:**
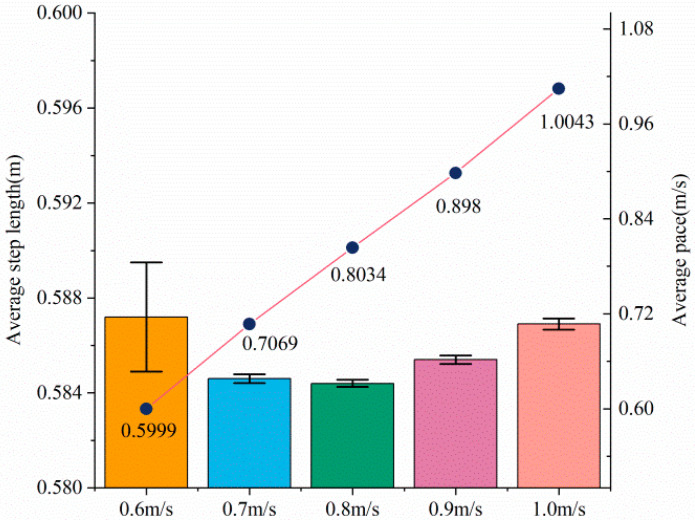
Experimental results of step size and pace.

**Figure 21 sensors-23-05311-f021:**
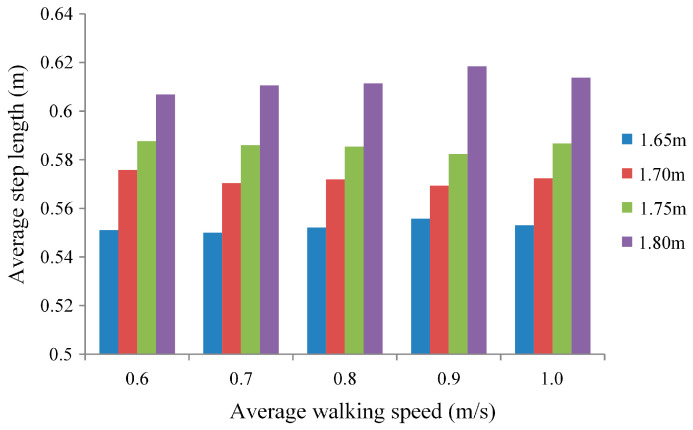
Relationship between step size and height.

**Table 1 sensors-23-05311-t001:** LLRER simulation model parameters.

Projects	Value	Projects	Value
Total body weight	70 kg	Exoskeleton unilateral thigh	6 kg
Human unilateral thigh	7 kg	Exoskeleton unilateral calf	5 kg
Human unilateral calf	4.41 kg	Thigh linkage length	0.48 m
Total exoskeleton weight	22 kg	Calf linkage length	0.35 m

**Table 2 sensors-23-05311-t002:** Single gait cycle knee HEI torque data.

Joint	Controller	Absolute Average (Nm)	Standard Deviation (Nm)	Maximum (Nm)	Minimum (Nm)
HIP	Position	6.4603	8.2309	15.6984	−14.7982
HIP	Impedance	0.8743	0.9622	2.0702	−0.7719
HIP	SCVV	1.1912	2.1218	4.1338	−3.0231
Knee	Position	8.4505	11.3968	26.4005	−15.2920
Knee	Impedance	1.1877	1.3801	2.0702	−0.7719
Knee	SCVV	3.0487	2.1289	2.2312	−3.0231

## Data Availability

Not applicable.

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
