# Peer review of "A Self-Coordinating Controller with Balance-Guiding Ability for Lower-Limb Rehabilitation Exoskeleton Robot"

_sensors, 2023, doi:10.3390/s23115311_

Round 1

Reviewer 1 Report

The text has some hyphenation issues, such as "de-vices" on line 34,
"de-pendent" on line 37, "inter-acts" on line 53, "re-al" on line 83 and
"
con-figuration" on line 110.

In section 4.1, line 514, it is informed that "The experimental procedure
involving human subjects", but it does not inform how many individuals
participated in the experiment and this is an important data.

The text between lines 524 and 531 is confusing to understand, because of
the way the punctuation symbol "colon" was used at the end of line 525.
What you would expect there would be a list, describing the design of the
experiment. I suggest turning the paragraph between these lines into an unordered list
(as seen on the line). If each step described is already in the order that the experiment should
be done, I suggest using the ordered
list.

The phrase "The small experimenter's own influence on the experimental
results, during the experiment, did not inform the experimenter that the
control parameters were changed", which is between lines 529 and 531 is
confused by the repeated use of variants of the noun experiment. The same problem with using the "colon" symbol is found between lines
597 and 601. Again, a list would be a good option here. Also, I
would rethink the use of the "colon" symbol at the end of line 601.

In conclusion, there could be some description of possible future work,
mainly involving a more detailed experiment with the participation of
human subjects.

Reviewer 2 Report

Paper is well organized and the topic presented try to correct the direction through self-coordination, synthesizes a reasonable reference velocity vector, and provides balanced guiding for the patient.

Author Response

I would like to express my sincere gratitude to the reviewers for taking the time out of their busy schedules to review my paper. I am truly grateful for their recognition of my work.

Reviewer 3 Report

The paper presents a self-coordinating controller with balance guiding ability for lower limb rehabilitation exoskeleton robot. The paper is well-written and presents interesting results. But some aspects may be improved:

1. Conclusions are general and contain only the summary of the paper. It should contain a discussion about the limitations and disadvantages of the presented approach as well as directions for future work.

2. Why experiments were performed for men of similar similar sizes? It would be interesting to compare results obtained for men of different heights and weights.  
